# Programmable DNA pyrimidine base editing via engineered uracil-DNA glycosylase

Zongyi Yi [1,4], Xiaoxue Zhang[2,4], Xiaoxu Wei[1,3,4], Jiayi Li[1,3,4], Jiwu Ren[2,3], Xue Zhang[2], Yike Zhang[1,3], Huixian Tang[1,3], Xiwen Chang[2,3], Ying Yu[1] & Wensheng Wei [1,2,3] ✉

DNA base editing technologies predominantly utilize engineered deaminases, limiting their ability to edit thymine and guanine directly. In this study, we successfully achieve base editing of both cytidine and thymine by leveraging the translesion DNA synthesis pathway through the engineering of uracil-DNA glycosylase (UNG). Employing structure-based rational design, exploration of homologous proteins, and mutation screening, we identify a *Deinococcus radiodurans* UNG mutant capable of effectively editing thymine. When fused with the nickase Cas9, the engineered DrUNG protein facilitates efficient thymine base editing at endogenous sites, achieving editing efficiencies up to 55% without enrichment and exhibiting minimal cellular toxicity. This thymine base editor (TBE) exhibits high editing specificity and significantly restores IDUA enzyme activity in cells derived from patients with Hurler syndrome. TBEs represent efficient, specific, and low-toxicity approaches to base editing with potential applications in treating relevant diseases.

The advancement of base editing tools holds significant promise for disease treatment, like repairing single-gene genetic diseases caused by SNPs[1]. Currently, there are developed cytosine base editors and adenine base editors. The deaminase in these tools transforms cytosine (C) or adenine (A) into uracil or inosine, which are subsequently recognized as thymine (T) and guanine (G), facilitating C-to-T and A-to-G base changes[2–4]. Building on this, the generation of apurinic/apyrimidinic sites (AP sites) occurs by cutting the intermediate products uracil or inosine with DNA glycosylase. Subsequent translesion DNA synthesis (TLS) incorporates other bases opposite the AP site, leading to the development of CGBE[5,6] and AYBE[7,8]. These base editing tools depend on deaminase enzymes, which restrict their ability to edit T and G. Recently, Tong et al. engineered N-methylpurine DNA glycosylase protein (MPG) to achieve guanine base editing[9]. The pursuit of base editors enabling T-to-C, T-to-G, and T-to-A mutations remains crucial for addressing a substantial number of point mutations, accounting for up to 70% of pathogenic human SNPs[3]. Consequently, the development of base editors for T-to-C, T-to-G, and T-to-A is of

great significance. Given the structural similarities among uracil, cytosine, and thymine, there exists an opportunity to engineer the active site of uracil-DNA glycosylase (UNG)[10] to enable the editing of both cytosine and thymine.

In this work, we achieve base editing of both T and C through a combination of structure-based rational design, exploration of homologous proteins, and mutation screening on *Deinococcus radiodurans* uracil-DNA glycosylase (UNG).

## Results

### Engineering of human UNG enables cytosine and thymine base editing

The active site pocket of UNG has primarily evolved to excise uracil (U). Despite the structural similarities among the pyrimidines, cytosine, thymine and uracil exhibit distinct differences, particularly the methyl group at the 5th position of thymine[11] (Fig. 1c). Based on structural analysis, amino acids G143-D145, Y147, and N204 play a role in influencing the size of its active site pocket in human UNG (hUNG)[12]

[1]Biomedical Pioneering Innovation Center, Peking-Tsinghua Center for Life Sciences, Peking University Genome Editing Research Center, State Key Laboratory of Protein and Plant Gene Research, School of Life Sciences, Peking University, Beijing, People's Republic of China. [2]Changping Laboratory, Beijing, People's Republic of China. [3]Academy for Advanced Interdisciplinary Studies, Peking University, Beijing, People's Republic of China. [4]These authors contributed equally: Zongyi Yi, Xiaoxue Zhang, Xiaoxu Wei, Jiayi Li. ✉e-mail: wswei@pku.edu.cn

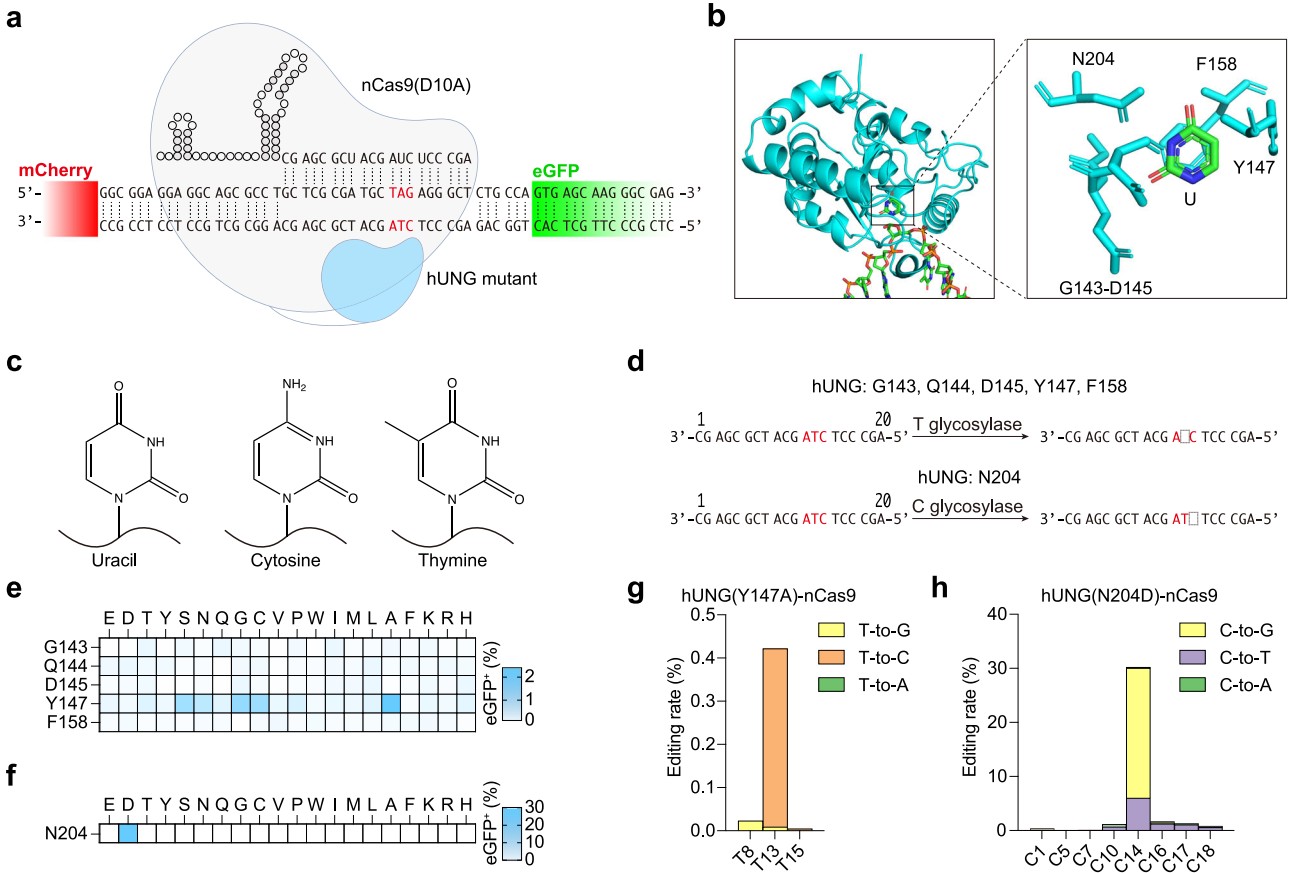

**Fig. 1 | Achieving base editing by excising bases using DNA glycosylase.**
**a** Schematic diagram of nCas9(D10A)-hUNG mutant targeting reporter system.
**b** The crystal structure and key amino acids responsible for hUNG recognizing and cleaving uracil bases (PDB: 1SSP). **c** The chemical structures of uracil, cytosine, and thymine bases. **d** Screen hUNG mutants that can specifically excise thymine and cytosine to produce AP sites using a reporter system. **e** The eGFP-positive ratio of

hUNG key amino acid saturation mutations for thymine excision. **f** The eGFP-positive ratio of hUNG key amino acid saturation mutations for cytosine excision. **g** The editing rate of hUNG(Y147A)-nCas9(D10A) targeting reporter system. **h** The editing rate of hUNG(N204D)-nCas9(D10A) targeting reporter system. **e–h** Data are presented as mean values of $n = 3$ independent biological replicates. Source data are provided as a Source Data file.

(Fig. 1b). Therefore, there is a possibility to introduce mutations in these amino acids, enabling the entry of C and T into the active site pocket and facilitating programmable C or T base editing (Fig. 1d).

To achieve this goal, we conducted saturation mutagenesis on several amino acids, including G143-D145, Y147, and F158, which could affect the entry of thymine into the active site pocket. Notably, Y147 was identified as hindering the entry of the methyl group at the 5th position of T. N204 was found to affect the entry of cytosine into the active site pocket. By fusing the mutated form of hUNG to the N-terminal of nCas9 (nickase Cas9, D10A) and targeting a previously developed premature stop codon reporter system[13,14] (Fig. 1a), we assessed editing efficiency based on the eGFP-positive ratio when UNG converted stop codons to normal codons. The results revealed that mutating Y147 of hUNG to amino acids with smaller side chains, such as A, C, G, or S, led to a detectable but low eGFP-positive ratio (Fig. 1e). This suggests that amino acids with smaller side chains may reduce obstacles for thymine to enter the active site pocket. The proportion of eGFP-positive cells reached 30% only when N204 mutates to D (Fig. 1f). These results are consistent with previous findings that the replacement of N147 by A or N204 by D results in hUNG shows thymine-DNA glycosylase activity or cytosine-DNA glycosylase activity, respectively[15]. In theory, the N204D mutation is expected to result in hydrogen bonding between the carboxylate of asparagine and the 4-amino group of cytosine, potentially enabling the editing of cytosine.

Through targeted sequencing of eGFP-positive HEK293T cells that transfected with Y147A and N204D mutants, the Y147A mutant exhibited 0.4% editing efficiency, mainly resulting in T-to-C conversion (Fig. 1g), while the N204D mutant showed the highest editing efficiency at 30%, predominantly resulting in C-to-G or C-to-T conversion (Fig. 1h). These editing results may be related to the endogenous translesion DNA synthesis pathway. Due to the different polymerases inserting various bases opposite an AP site (apurinic/apyrimidinic site) during DNA synthesis, the editing outcomes may exhibit preferences. Collectively, these results indicate that direct editing of C or T can be achieved through the engineering of human UNG.

## Screening UNG and its mutants from other species for more efficient thymine base editing

To further enhance thymine excision efficiency, we redirected our attention to UNG derived from various species. Employing protein sequence alignment, we specifically selected UNG from *Escherichia coli* (Ec)[16], *Deinococcus radiodurans* (Dr)[17], *Human Herpesvirus 1* (HHV1)[18], and *Vaccinia virus* (VACV)[19]. Utilizing homologous protein sequence alignment, we identified the corresponding amino acids in these UNG variants analogous to N204 and Y147 of hUNG. We then introduced mutations to these amino acids, changing them to D and A, respectively, to assess their efficacy in excising cytidine and thymine. We found that HHV1_UNG(N147D) achieved higher cytidine editing efficiency than hUNG(N204D), while EcUNG(N123D) was comparable to

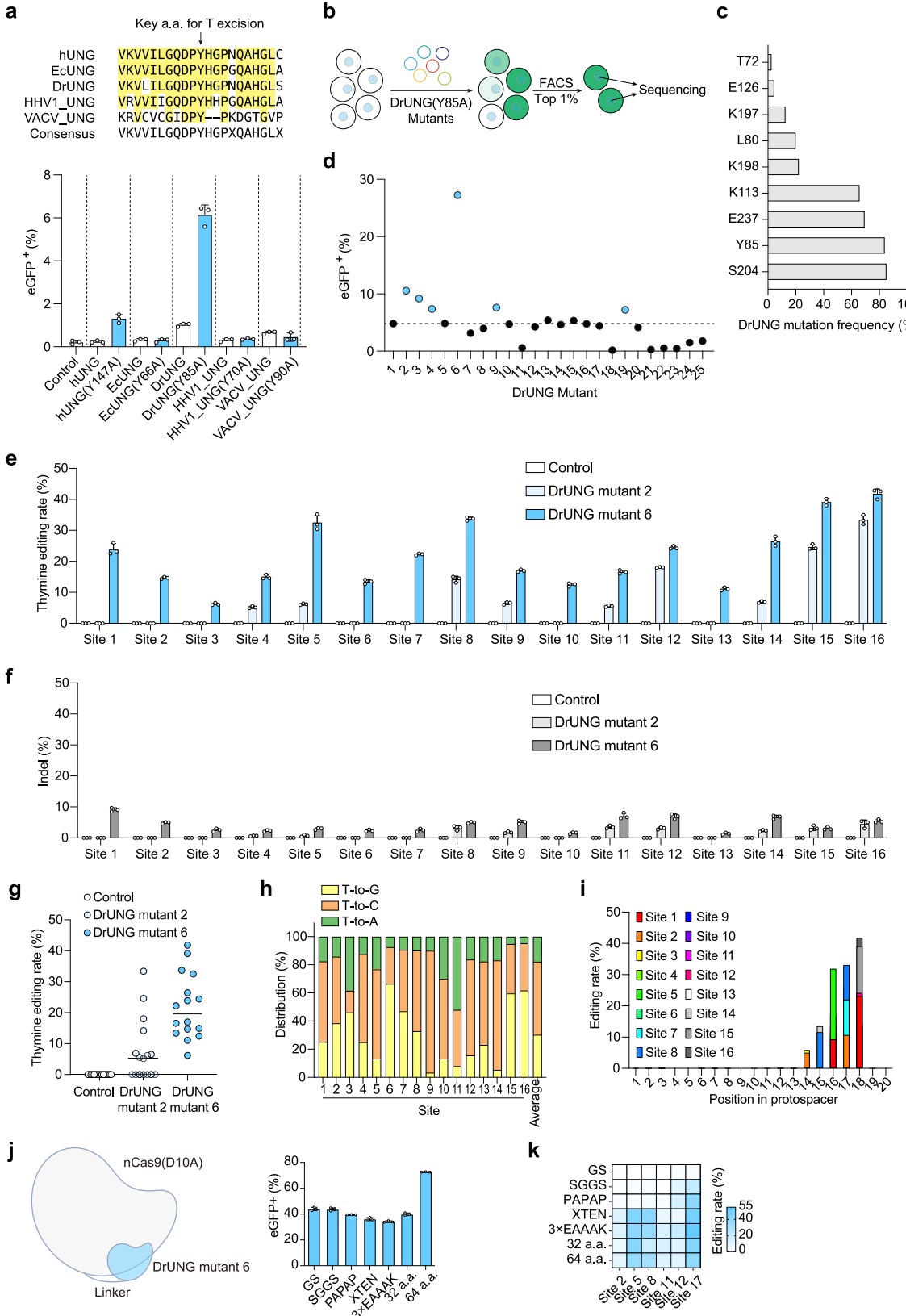

hUNG(N204D) (Supplementary Fig. 1). Regarding thymine editing, we found that wild-type DrUNG has comparable editing efficiency to hUNG(Y147A), while DrUNG(Y85A) exhibited the highest efficiency, nearly five times greater than that of hUNG(Y147A) (Fig. 2a). Despite this notable enhancement, there is still room for improvement in thymine editing efficiency when compared to cytidine editing.

Consequently, we attempted further optimizations for the thymine base editor.

To enhance the activity of DrUNG mutants, we generated a library of DrUNG(Y85A) mutants. Subsequently, we utilized the DrUNG(Y85A) mutants-nCas9(D10A) library to target the reporter system and sorted the top 1% of eGFP-positive cells through FACS, and then amplified the

**Fig. 2 | Improve the efficiency of pyrimidine base editing via searching more efficient homologous proteins and introducing mutations. a** Top, search for more efficient homologous proteins for thymine excision. Bottom, the ratio of eGFP-positive cells illuminated by UNG and its mutants from different species. **b** Flowchart for screening DrUNG(Y85A) mutants. **c** The enriched amino acid site in DrUNG(Y85A) mutants screening. **d** The eGFP-positive ratio of different DrUNG(Y85A) mutants. **e** Thymine base editing rate of DrUNG mutants 2 and 6 at 16 endogenous sites. **f** The indel of DrUNG mutants 2 and 6 at 16 endogenous sites. **g** Comparison of thymine base editing efficiency of DrUNG mutants 2 and 6 at 16 endogenous sites. **h** Distribution of thymine editing results at 16 endogenous sites of DrUNG mutant 6. **i** The position distribution of thymine editing results of DrUNG mutant 6 relative to spacer at 16 endogenous sites. **j, k** The editing efficiency of the reporter system (**j**) and endogenous site (**k**) when using different linkers between nCas9 and DrUNG mutant 6. **a, e, f, h, i, j** and **k** Data are presented as mean ± s.d. of *n* = 3 independent biological replicates. **g** Data are presented as the mean editing rate of 16 sites. Each dot represents the mean editing rate of the target site. Source data are provided as a Source Data file.

sequences of DrUNG(Y85A) mutants (Fig. 2b). We found that the mutations of the screened mutants were enriched in 9 amino acids, mainly S204, Y85, E237 and K113 (Fig. 2c). By validating 25 mutants that artificially combined these mutations (DrUNG mutant 1 contains only the Y85A mutation and serves as a control and the mutations contained in these mutants are listed in Supplementary Data 2), we found that the 6th mutant variant exhibited the highest editing efficiency, followed by the 2nd mutant variant (Fig. 2d). These two variants were designated DrUNG mutant 6 and DrUNG mutant 2, respectively. DrUNG mutant 6 encompasses mutations L80V, Y85A, K113E, S204A and E237Q. DrUNG mutant 2 includes mutations Y85A, K113E, and S204A. Compared to DrUNG(Y85A), both DrUNG mutant 6 and DrUNG mutant 2 demonstrated a 5.4-fold and 2.2-fold increase in editing efficiency, respectively. Additionally, when compared to hUNG(Y147A), these two mutants exhibited a 27-fold and 11-fold increase in editing efficiency (Fig. 2d).

Subsequently, we utilized these two DrUNG mutants to target 16 endogenous sites in HEK293T cells. Notably, DrUNG mutant 6 exhibited effective editing at all targeted sites, whereas DrUNG mutant 2 exhibited effective editing at 9 sites. The peak editing efficiency for DrUNG mutant 6 reached 40%, while for DrUNG mutant 2, it reached 33% (Fig. 2e). Concurrent with targeted editing, the thymine base editor induced indels at the targeted sites, with the highest indel rate being 10% (Fig. 2f). Indels caused by the thymine base editor based on DrUNG, 76% are deletions and 24% are insertions (Supplementary Fig. 2). This may be attributed to the double-strand DNA activity of the UNG mutants. When thymines on both DNA strands are excised, it can easily cause double-strand breaks (DSBs). This aspect requires further optimization in future research.

In summary, the average editing efficiency of DrUNG mutant 6-mediated base editors was 20%, which is three times that of DrUNG mutant 2 (Fig. 2g). Examining the editing outcomes of the thymine base editor, we found that approximately 52% of thymine converts to cytosine, around 30% transforms into guanine and approximately 18% changes to adenine on average (Fig. 2h). Notably, there are variations in the distribution of T conversion among different sites. Upon analyzing the editing positions, it became apparent that the majority of edits occurred at positions 14 to 18 away from the NGG PAM, exhibiting a similar distribution pattern to other types of CRISPR-based base editors (Fig. 2i). For cytosine base editing based on HHV1_UNG(N147D), cytosine is mainly converted into guanine, followed by thymine and adenine (Supplementary Fig. 3a). Moreover, its editing window is mainly located at bases 13 to 19 from the PAM position (Supplementary Fig. 3b). In addition, we tried to use different linkers (Supplementary Data 3) between DrUNG mutant 6 and nCas9(D10A), and found that using a 64-amino acid linker on the reporter system was the most efficient, while using GS, SGGS, PAPAP, XTEN, 3×EAAAK and 32-amino acid linkers showed similar editing efficiency (Fig. 2j). Using XTEN, 3×EAAAK, 32-amino acid and 64-amino acid linkers show higher editing efficiency compared to GS, SGGS and PAPAP linkers on multiple endogenous sites and the editing efficiency is as high as 55% (Fig. 2k). Based on this, we recommend using a longer linker to achieve higher editing efficiency. We named the base editors based on DrUNG mutant 6 as thymine base editors (TBEs). TBEs show strong editing activity and low indel levels in a variety of cell lines on the reporter system,

including human cell line HCT116, mouse cell lines Neuro-2a and NIH3T3, monkey cell line Cos-7, among which Neuro-2a and NIH3T3 cell lines achieved approximately 70% editing efficiency on reporter system (Supplementary Fig. 4a, b). To conclude, the thymine base editors we developed exhibit effective editing in multiple cell lines.

## TBEs show high editing specificity at both the genome and transcriptome levels
We evaluated the editing specificity of TBEs on a genome-wide and transcriptome-wide scale. Through genome-wide high-throughput sequencing detection, we found that the TBEs transfection group showed some off-targets compared with the control group (Fig. 3a, Supplementary Fig. 5a, and Supplementary Data 4). We took out the 50 bp upstream and downstream of these off-target sites and found there are no potential sgRNA binding sites. Furthermore, we used Cas-OFFinder to predict the potential Cas9-dependent off-target sites for four sgRNAs across the genome. We selected the top 10 highest-scoring off-target sites for sequencing and found that none of these sites were edited (Fig. 3b). This suggests that the off-target effects we observed are likely random and are comparable to background editing (Fig. 3a). Furthermore, through a Cas9-independent off-target analysis using an orthogonal R-loop assay[20] (Fig. 3c), we detected no significant off-target effects for TBEs, while ABEs exhibited notable off-target activity compared to the untreated group, with the highest off-target editing efficiency reaching 26.7% (Fig. 3d). In addition, we observed no significant off-target effects at the transcriptome level (Fig. 3e and Supplementary Fig. 5b).

In summary, TBE does not cause severe off-target editing at either the genome or transcriptome levels, indicating that TBE is a relatively safer option for thymine-based editing.

## TBE exhibits low or even negligible cytotoxicity
Recently, two research groups have published thymine base editors utilizing hUNG mutants[21,22]. We compared their hUNG mutants with our DrUNG mutant 6 and found that DrUNG mutant 6 exhibits higher editing efficiency, both in a reporter system and at multiple endogenous sites (Fig. 4a, b). An overall comparison of editing efficiency at these 32 endogenous sites revealed that DrUNG mutant 6 significantly outperformed the hUNG mutants (Fig. 4c). After transfection with HEK293T cells, it was observed under a microscope that cells transfected with DrUNG mutant 6 appeared healthier than those transfected with the other three hUNG mutants three days post-transfection, resembling the untreated cells (Fig. 4d, f). In addition, the growth rate of cells transfected with DrUNG mutant 6 was superior to those transfected with the three hUNG mutants, closely mirroring that of untreated cells (Fig. 4e). This indicates that DrUNG mutant 6 has lower cytotoxicity than hUNG mutants.

## Incorporating polymerase can improve the purity of TBE editing
We then attempted to overexpress Polymerase η, a translesion synthesis (TLS) polymerase reported to enhance adenine editing purity[8], within the TBE system. Remarkably, this modification led to a 1.1 to 6-fold increase in T-to-A conversion rates. This demonstrates that co-expressing polymerase can significantly refine the specificity of thymine conversion into targeted bases (Supplementary Fig. 6).

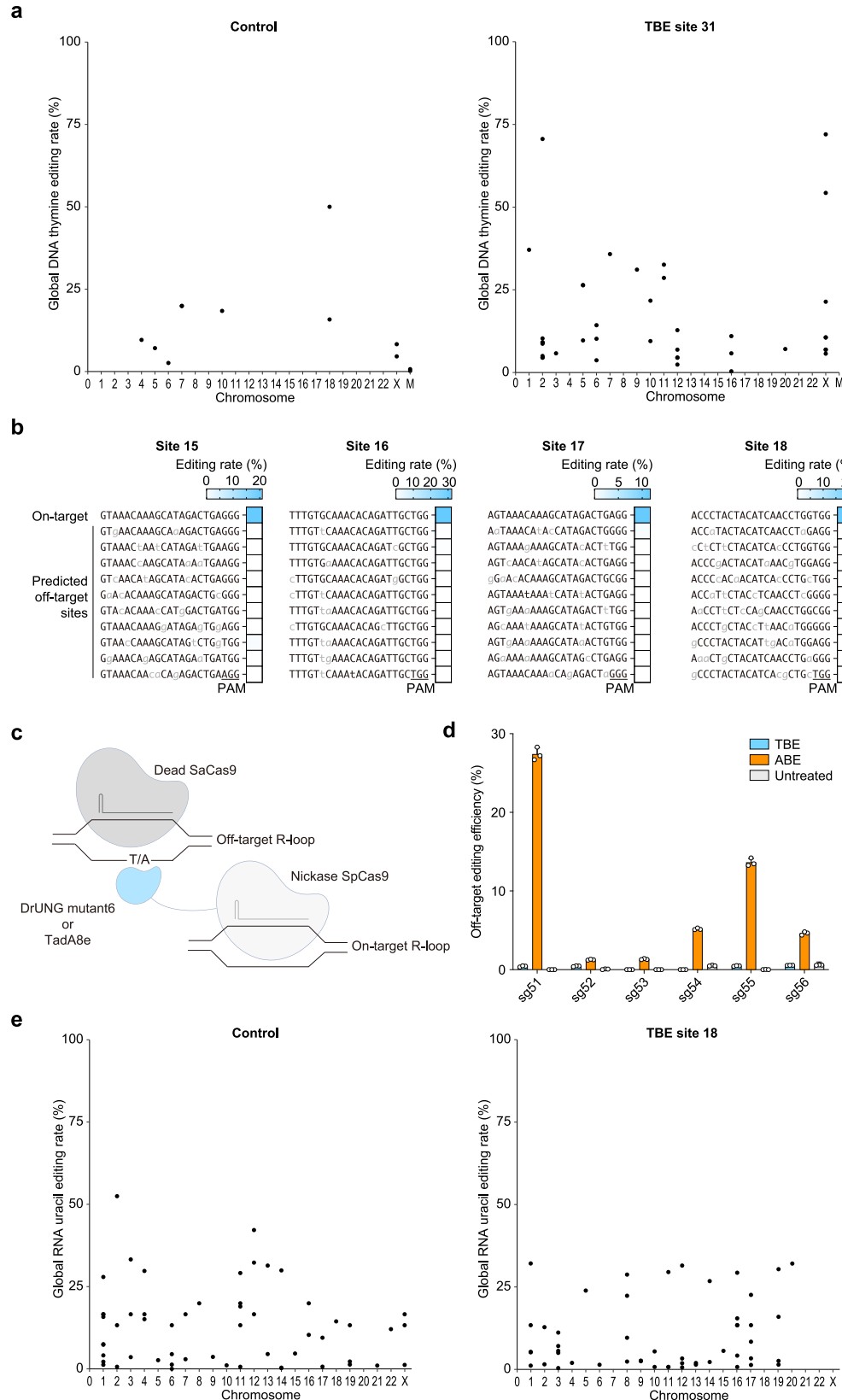

**Fig. 3 | Off-target assessment of TBEs across the whole genome and transcriptome. a** Genome-wide off-target analysis of TBE at site 31, with a sample transfected with an eGFP-expressing plasmid serving as a control. **b** Editing efficiency at the top 10 potential Cas9-dependent off-target sites predicted by Cas-OFFinder at sites 15, 16, 17, and 18. **c** Schematic diagram of Cas9-independent off-target analysis by an orthogonal R-loop assay. **d** Editing efficiency of TBE and ABE at six Cas9-independent off-target sites. **e** Transcriptome-wide off-target analysis of TBE at site 18, with a control sample transfected with an eGFP-expressing plasmid. Panels **a**, **b**, and **e** present the mean values from three biological replicates, while panel **d** shows data as mean ± s.d. from $n = 3$ independent biological replicates. Source data are provided as a Source Data file.

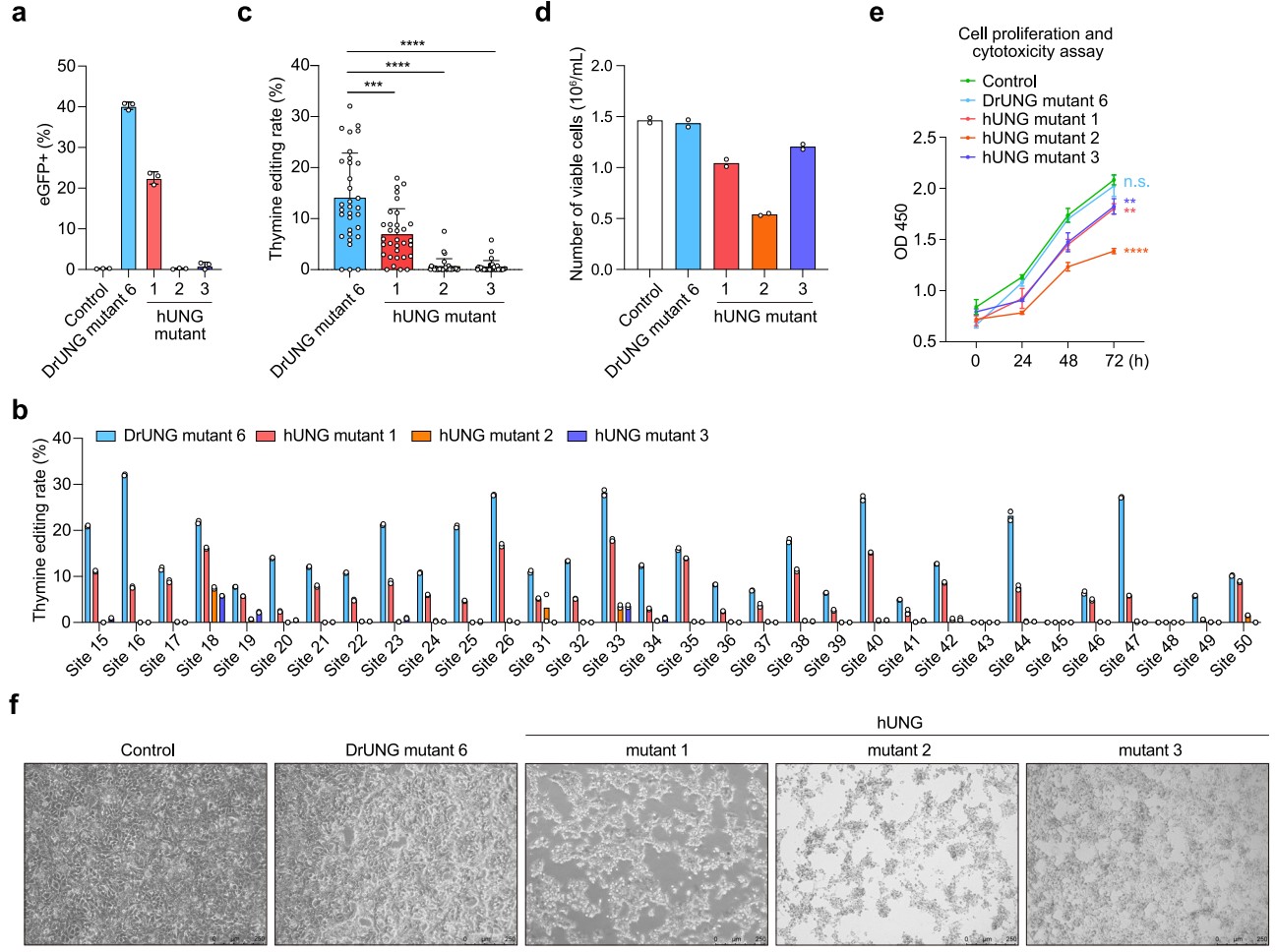

**Fig. 4 | DrUNG mutant 6 shows higher editing efficiency and lower cytotoxicity than hUNG mutants. a** The eGFP-positive ratio of the reporter system using nCas9(D10A) fused DrUNG-mutant 6, hUNG mutant 1, 2, and 3. **b** Editing efficiency at multiple endogenous sites with sgRNA and nCas9(D10A) fused DrUNG-mutant 6, hUNG mutant 1, 2 and 3. **c** Compare editing efficiency between different UNG mutants. **d** Statistics of the number of viable cells after base editor treatment. **e** Proliferation and cytotoxicity of cells transfected with sgRNA and nCas9(D10A) fused DrUNG-mutant 6, hUNG mutant 1, 2 and 3. **f** Cell status transfected with sgRNA and nCas9(D10A) fused DrUNG-mutant 6, hUNG mutant 1, 2 and 3 (scale bar, 250 μm). **a, e** Data are presented as mean ± s.d. of $n = 3$ independent biological replicates. **b, d** Data are presented as mean of $n = 2$ independent biological replicates. **c** Data are presented as mean ± s.d. of 32 target sites. Student's $t$-test was used for all statistical comparisons between different groups in **c** and **e**. $P$-value = 1.74E-04 for DrUNG mutant6 vs. hUNG mutant1; $P$-value = 5.37E-12 for DrUNG mutant6 vs. hUNG mutant2; $P$-value = 3.65E-12 for DrUNG mutant6 vs. hUNG mutant3 in **c**. $P$-value = 4.11E-01 for DrUNG mutant6 vs. Control; $P$-value = 2.21E-03 for hUNG mutant1 vs. Control; $P$-value = 2.62E-05 for hUNG mutant2 vs. Control; $P$-value = 8.27E-03 for hUNG mutant3 vs. Control in **e**. **f** Three times the experiment was repeated with similar results. Source data are provided as a Source Data file.

## Significant restoration of Hurler syndrome disease cell phenotypes using TBE

TBE is a tool that can be used in the treatment of many mutation-related diseases. We first applied it to the treatment of Hurler syndrome. One cause of Hurler syndrome is due to the presence of a premature stop codon in exon 9 of the IDUA gene, which prevents the production of functional IDUA protein. We selected three different nickase Cas9 proteins (SpCas9, SpCas9-NG, and SpG-Cas9)[23] and designed their corresponding sgRNA. By transfecting cells of the previously reported Hurler syndrome disease premature stop codon reporter system[24] (Fig. 5a), we found that all three nickase Cas9 proteins fused to DrUNG mutant 6 can achieve thymine base editing, among which SpCas9 protein has the highest editing efficiency (Fig. 5b). Subsequently, we co-transfected the mRNA of this TBE version and sgRNA into GM06214(IDUA^W402X) cells derived from a patient with Hurler syndrome, which contain the IDUA^W402X mutation (Fig. 5c), achieving an editing efficiency of approximately 25% at the targeted site and significantly restoring the IDUA catalytic activity of

the cells (Fig. 5d, e). This shows that TBE has the potential for disease treatment.

## Discussion

In this study, we have introduced a base editing tool that operates independently of deaminases. By using DNA glycosylase to selectively remove thymine or cytosine, generating an AP site, and inducing nicking on the complementary strand with nCas9, we achieve base editing of cytosine or thymine during translesion DNA synthesis by incorporating alternative bases opposite the AP site. Our experiments have revealed a preference for incorporating G opposite the AP site after thymine removal, resulting in T-to-C base editing. A smaller fraction of incorporations involves C or T, allowing for T-to-G and T-to-A base conversions (Fig. 2). In the context of human disease-related point mutations, our base editor holds the potential to correct up to 70% of these mutations. Notably, existing base editing tools relying on deaminases often exhibit significant off-target effects on RNA, posing a clear drawback for disease treatment. Our tool, circumventing the

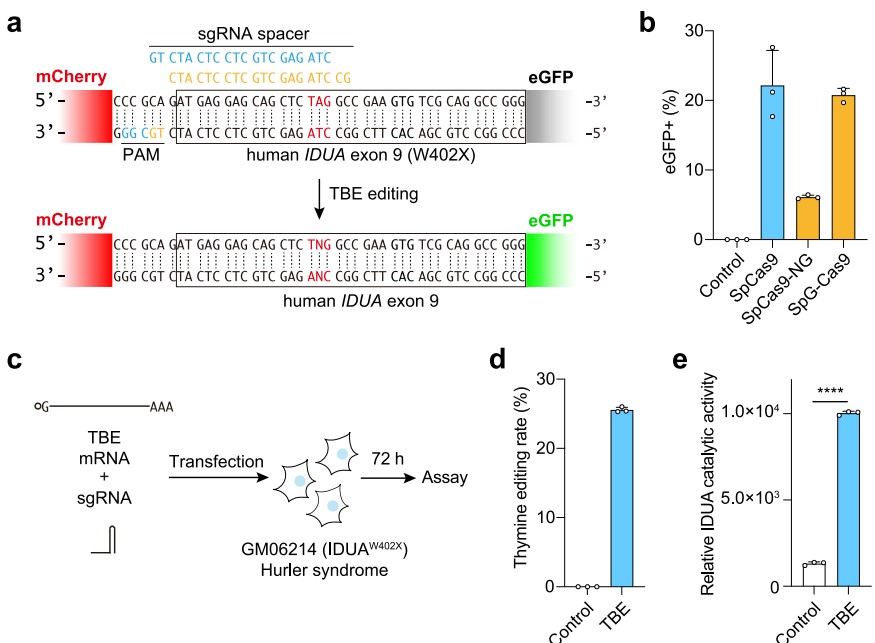

**Fig. 5 | Significant restoration of Hurler syndrome disease cell phenotype using TBE. a** Diagram illustrating the human Hurler syndrome reporter system and sgRNA design for TBE. **b** Editing efficiency of SpCas9, SpCas9-NG, and SpG-Cas9 respectively fused to DrUNG mutant 6 and co-transfected with the corresponding sgRNA in IDUA reporter system. **c** Schematic diagram of co-transfection of mRNA of the fusion protein of SpCas9 and DrUNG mutant 6 and sgRNA into GM06214(IDUA^W402X) cells derived from a patient with Hurler syndrome, which contain the IDUA^W402X mutation. **d** Editing efficiency of on-target in GM06214 cells using TBE. **e** Relative IDUA catalytic activity of GM06214 cells after transfection with TBE. **b**, **d**, and **e** Data are presented as mean ± s.d. of $n = 3$ independent biological replicates. Student's $t$-test (two-tailed) was used for statistical comparisons between different groups and $P$-value = 6.01E-08 for TBE vs. Control in **e**. Source data are provided as a Source Data file.

need for deaminases that cause global RNA off-target, provides a more precise approach to base editing for therapeutic applications (Fig. 3 and Supplementary Fig. 5). In this study, we tried to use it for the treatment of Hurler syndrome disease. We co-transfected the mRNA of DrUNG mutant 6 and nickase SpCas9 fusion protein and sgRNA into Hurler syndrome disease cells GM06214(IDUA^W402X), achieving about 25% editing efficiency and significantly restoring the catalytic activity of IDUA (Fig. 5).

Through guided protein structure-based engineering, we identified mutant variants N204D and Y147A of the hUNG protein capable of specifically excising cytosine and thymine (Fig. 1). Building on this discovery, we identified HHV1_UNG(N147D) and DrUNG(Y85A) through homologous protein searches, each demonstrating more efficient excision of cytosine and thymine, respectively. The activity of DrUNG(Y85A) surpasses that of hUNG(Y147A) by a factor of five, and through further mutations, we enhanced the activity of DrUNG(Y85A) by approximately six-fold (Fig. 2). Consequently, we have developed an effective thymine base editing tool, with the editing window situated at positions 14-18 of the spacer. Engineered DrUNG-based thymine base editors show low off-targets across the entire genome. In addition, it has no obvious off-target sites at the RNA level. Compared to two recently published hUNG-based thymine base editors[21,22], the DrUNG-based thymine base editor exhibits not only higher editing efficiency but also minimal cellular toxicity. In contrast, the hUNG editors exhibit significant cytotoxic effects (Fig. 3). However, it is important to acknowledge that this base editing tool currently faces some limitations, including a relatively high indel rate, necessitating further optimization in the future. Additionally, the combination of different DNA polymerases exhibiting specific preferences can be explored to achieve the incorporation of specific base[8,9] (Supplementary Fig. 6). In summary, TBEs offer efficient, precise, and low-toxicity methods to base editing, presenting the potential for therapeutic applications in related diseases.

## Methods

### Plasmid construction

PCR was performed using PrimeSTAR GXL Premix (TaKaRa, #R051B) or Q5 Hot Start High-Fidelity 2X Master Mix (NEB, #M0492). Wild-type hUNG, EcUNG, DrUNG, HHV1_UNG, VACV_UNG and its variants, Cas9, and other genes were synthesized as gene blocks and codon optimized for mammalian expression (Tsingke Biotechnology Co., Ltd.) (sequence is shown in (Supplementary Data 1). We constructed the mutation UNG fragment into the pCMV vector by Gibson assembly using Gibson Assembly Master Mix (NEB, #M5510). Individual sgRNA oligos (sgRNA sequence is shown in Supplementary Data 5) were synthesized and cloned into pCG-2.0 sgRNA-expressing vector through Golden-Gate assembly kit (NEB, #E1602). Ligated plasmids were transformed into Trans1-T1 chemically competent cells (Trans-Gene Biotech, #CD501) and subjected to Sanger sequencing to analyze the identity of the constructs (Rui Biotech). Final plasmids were prepared (TianGen) for cell transfection.

### Preparation of DrUNG mutant library

The library of mutant DrUNG-Cas9 plasmids was constructed through the assembly of two fragments. Fragment 1 comprises a mutated DrUNG gene with NotI/BamHI restriction enzyme sites, which was generated via error-prone PCR using 1 μl of Mutazyme II DNA polymerase (Agilent GeneMorph II Random Mutagenesis Kit, #200550), 5 μl of 10×Mutazyme II reaction buffer, and 1 μl of 40 mM dNTP mix (final concentration of 200 μM each) in a 50 μl PCR reaction with approximately 0.2 ng/μl template derived from the DrUNG(Y85A) template. Fragment 2 comprises the remaining portion of the base editor plasmid, containing nCas9 from CGBE1 (plasmid PRZ3885 purchased from Addgene), which was double digested with NotI/BamHI enzymes (New England Biolabs). Both fragments were purified via agarose gel electrophoresis using the Zymoclean Gel DNA Recovery Kit (Zymo Research, #D4002). Subsequently, the library of DrUNG mutant

plasmids was assembled using the Gibson assembly method in a 20 µl reaction.

## Production and purification of mRNA

The production of mRNAs referred to the manufacturer's instructions. Briefly, we produced the mRNAs using the commercial HiScribe™ T7 High Yield RNA Synthesis Kit (NEB, #E2040S) according to the manufacturer's instructions with the linearized plasmids containing the T7 promotor, UNG mutation, nCas9(D10A) and -225-nt polyA elements. Final IVT products were column purified and concentrated with the RNA Clean & Concentrator Kit (ZYMO, #R1018).

## Electroporation in primary cells

Electroporation in primary cells. For mRNA electroporation in GM06214 cells, 1.5 µg sgRNA and 4.5 µg DrUNG-nCas9 mRNA were electroporated with Nucleofector 2b Device (Lonza) and Basic Nucleofector Kit (Lonza, #VPI-1002), and the electroporation program was U-012. Then the cells were transferred to a warm culture medium for the following assays. Cells were collected after 72 h of transfection and detected the IDUA catalytic activity.

## IDUA catalytic activity assay

The gathered cell pellet was resuspended and lysed with 28 µl 0.5% Triton X-100 in 1× PBS buffer on ice for 30 min. Then 25 µl of the cell lysis was added to 25 µl 190 µM 4-methylumbelliferyl-α-l-iduronidase substrate (Cayman, #2A-19543-500), which was dissolved in 0.4 M sodium formate buffer containing 0.2% Triton X-100, pH 3.5 and incubated for 17 h at 37 °C in the dark. The catalytic reaction was quenched by adding 200 µl 0.5 M NaOH/Glycine buffer, pH 10.3, and then centrifuged for 2 min at 4 °C. The supernatant was transferred to a 96-well plate, and fluorescence was measured at 365 nm excitation wavelength and 450 nm emission wavelength with Infinite M200 reader (TECAN).

## Cell line construction

To create the dual-fluorescence reporter, mCherry and eGFP coding sequences (the ATG start codon of eGFP was deleted) were PCR amplified and digested using BsmBI (Thermo Fisher Scientific, no. ER0452) before being subjected to T4 DNA ligase (NEB, #M0202L)-mediated ligation with 3×GGGGS linkers. The ligation product was subsequently inserted into the pLenti-CMV-MCS-PURO backbone. To construct stable reporter cell lines, reporter constructs (pLenti-CMV-MCS-PURO backbone) were cotransfected into HEK293T cells together with two viral packaging plasmids, pR8.74 and pVSVG. After 72 h, viral supernatant was collected and stored at −80 °C. HEK293T cells were infected with lentivirus, and then mCherry⁺ cells were sorted via fluorescence-activated cell sorting (FACS) and cultured to select a single clone cell line stably expressing a dual-fluorescence reporter system with no detectable eGFP background. The same sequence was also used in Reporter-1 which can be found on addgene (#180218).

## Cell culture and transfection

HEK293T (ATCC, CRL-3216) were obtained from C. Zhang's laboratory (Peking University). HCT116 (CCL-247), Neuro-2a (CCL-31), NIH3T3 (CRL-1658), and COS-7 (CRL-651) cell lines were maintained in W Wei's laboratory (Peking University). GM06241 (CTCC-001-0896) were purchased from Meisen. All the cells above and the dual-fluorescence reporter cells were cultured in Dulbecco's modified Eagle's medium (DMEM, Gibco, # C11995500BT) with 10% fetal bovine serum (Biological Industries) and penicillin/streptomycin (Sigma) at 37 °C with 5% $CO_2$. GM06214 were from MEISEN CELL and cultured in Dulbecco's modified Eagle's medium (DMEM, Gibco) with 15% fetal bovine serum (Biological Industries), 1% Non-Essential Amino Acid (NEAA) and penicillin/streptomycin (Sigma) at 37 °C with 5% $CO_2$. For lipofection, cells were plated in 12-well cell culture plates at a density that

approximately reached 70% after 20 h. Cells in each well were transfected with 2,250 ng of UNG mutation-nCas9(D10A) and 2250 ng of sgRNA using 9 µL of PEI (ProteinTech, #PR400001) or transfected with 2,500 ng of each UNG monomer mRNA using 5 µL of Lipofectamine MessengerMAX Reagent (Invitrogen, #LMRNA015). Cells were collected after 72 h of transfection. Genomic DNA was extracted using the DNeasy Blood & Tissue Kit (Qiagen, #69504) and stored at −20 °C.

## UNG variants screening by fluorescence-activated cell sorting (FACS) analysis

To assess UNG variants editing efficiency with the dual-fluorescence reporter system, HEK293T reporter cells were plated in 12-well cell culture plates. After 72 h of transfection, mCherry, BFP, and eGFP fluorescence were analyzed by flow cytometer. The mCherry signal served as a fluorescent selection marker for reporter-expressing cells, and the BFP signal served as a fluorescent selection marker for sgRNA-expressing cells. Percentages of eGFP⁺/mCherry⁺ cells were calculated as the readout for editing efficiency. FACS data were analyzed with FlowJo X (v.10.0.7). An Illustration of FACS shown in Supplementary Fig. 7.

## Cytotoxicity assay

HEK293T cells were seeded in 96-well plates (Corning) at $2×10^4$ cells per well in 200 µl of complete growth medium. 24 h after seeding, cells were transfected with 1 µl PEI (ProteinTech, #PR400001), 250 ng sgRNA, and 250 ng editors. 0 h, 24 h, 48 h, and 72 h after transfection, 20 µl CCK8 solution (DOJINDO, #CK04-05) was added to each well and the absorbance at 450 nm was measured after 1.5 h using a microplate reader.

## Targeted deep sequencing

Genomic sites of interest were amplified into fragments of approximately 200 bp from genomic DNA samples using PrimeSTAR GXL Premix (TaKaRa, #B051B) (the primer sequence is shown in Supplementary Data 5). PCR products were purified using DNA Clean & Concentrator-25 (Zymo Research, #D4006) for Sanger sequencing and targeted deep sequencing. Targeted deep sequencing libraries were prepared using the VAHTS Universal DNA Library Prep Kit for Illumina V3 (Vazyme, #ND607). Briefly, the PCR fragments were sequentially subjected to end repair, adapter ligation, and then PCR amplification. DNA purification in library preparation was performed using Agencourt Ampure XP beads (Beckman Coulter), and library amplification was performed using Q5U Hot Start High-Fidelity DNA Polymerase (NEB, #M0515) and VAHTS Multiplex Oligos Set 4/5 for Illumina (Vazyme, #N321). The final library was subjected to quantification using the Qubit dsDNA HS assay kit (Invitrogen) and sequenced using Illumina HiSeq X Ten.

## Genome-wide DNA off-target sequencing

We input 500 to 1000 ng of genomic DNA for library preparation using the VAHTS Universal Plus DNA Library Prep Kit for Illumina (Vazyme, #ND610). The process of library preparation was as follows: fragmentation, end preparation and dA-tailing, adapter ligation, and library amplification. Among them, 500 to 1000 ng of genomic DNA was fragmented with FEA enzyme mix at 37 °C for 10 min, and end repair and dA-tailing were simultaneously completed in the process. The final library was subjected to quantification using the Qubit dsDNA HS assay kit (Invitrogen, #Q32851) and fragment analyzer. All libraries were finally sequenced using Illumina HiSeq X Ten (Illumina).

## Transcriptome-wide off-target sequencing

HEK293T cells were transfected with either eGFP-expressing (Control) or TBEs-expressing and sgRNA-expressing plasmid. After 72 h post-transfection, RNA extraction was performed using Direct-zol RNA Miniprep Kits (Zymo Research, #R2052). Subsequently, RNA was

isolated using Ribo-off rRNA Depletion Kit (H/M/R) (Vazyme, #N406) and subjected to processing with the Universal V6 RNA-seq Library Prep Kit for Illumina (Vazyme, #NR604-01). The prepared samples underwent deep sequencing analysis utilizing the Illumina HiSeq X Ten platform.

### Cas9-dependent DNA off-target sequencing

Cas-OFFinder (CRISPR RGEN Tools (rgenome.net)) was used for the prediction of potential off-target sites of Cas9 RNA-guided endonucleases, the top 10 off-target sites were selected for validation. Ten off-target sites of each target site were amplified from genomic DNA prepared (Same as Targeted deep sequencing) and sequenced on the Hi-TOM NGS platform.

### Cas9-independent DNA off-target analysis

Human cell orthogonal R-loop assay was conducted to analyze Cas9-independent off-target in the TBE system. We co-transfected 12-well HEK293T cells with 1350 ng plasmids encoding a SpCas9-based TBE (DrUNG mutant 6) or ABE (TadA8e), 900 ng SpCas9 on-target guide RNA (sgRNA), 1,350 ng catalytically inactive Staphylococcus aureus Cas9 (dSaCas9) and 900 ng SaCas9 sgRNA targeting a genomic locus unrelated to the on-target site. All six SaCas9 sgRNA targeting genomic loci were reported before to evaluate the Thymine base editor off-target effect. Genomic sites of interest were amplified into fragments of approximately 200 bp from genomic DNA samples using PrimeSTAR GXL Premix (TaKaRa, #R051B) (primer sequence is shown in Supplementary Data 5) and the following steps were as same as targeted deep sequencing.

### Analysis of high-throughput sequencing data for targeted amplicon sequencing

For high-throughput sequencing data analysis, an index was generated using the targeted site sequences (upstream and downstream ~100 nt) of editing window-covered regions. The reads were aligned and quantified using BWA (v.0.7.10-r789). The BAM alignment files were then sorted with SAMtools (v1.1), and the editing sites were analyzed using REDitools (v.1.0.4)[25]. The parameters were as follows: -t 8 -U [AG] -n 0.0 -T 6-6 -e -d -u. All the significant base conversions within the targeted regions calculated by Fisher's exact test ($P$-value < 0.05) were considered edits made by the UNG variants. The mutations that appeared in the control and experimental groups simultaneously were considered to be due to single nucleotide polymorphisms.

### Analysis of nuclear genome off-target editing

Whole-genome sequencing reads underwent quality control using FastQC (v0.12.1), and adapters were removed using fastp (0.23.2). Following trimming, reads were aligned to the GRCh38-hg38 reference genome using bwa-mem2 (2.2.1) with default parameters. Subsequently, the Picard AddOrReplaceReadGroups (v2.25.5), MarkDuplicatesSpark, BaseRecalibratorSpark, and ApplyBQSRSpark were applied to add read group information, remove duplicates, and correct base quality. After preprocessing, GATK Mutect2 was utilized to identify somatic short variants. Variant calls were filtered based on the FilterMutectCalls criteria, excluding positions annotated as position, slippage, weak evidence, or low mapping quality. Additionally, mutations with a frequency exceeding 1% in control experiments were excluded. Only mutations at positions where the reference genome contained a T and the mutated allele was A/C/G were retained. To identify potential off-target genome editing events, stringent criteria were applied to mitigate high noise levels. Additional requirements for base quality and mapping quality were imposed based on quality control criteria. Only mutations with a high median base quality ($MBQ \geq 32$) and high mapping quality ($MMQ \geq 50$) were considered potential off-target editing sites. The Mann–Whitney $U$-test was employed to assess the significance of differences in mutation frequency between each experimental group and the control group ($P$-value < 0.1).

### Analysis of transcriptome off-target editing

The quality control of RNA-seq data was carried out as previously outlined. Alignments were executed using the two-pass mode of STAR (version 2.7.11a), and variant calling was conducted in accordance with the standard GATK pipeline. To ensure a high level of confidence in the variants identified from RNA-seq data, more stringent criteria were employed beyond the conventional control of median base and mapping quality, such as requiring a minimum depth of 50.

### Statistics and reproducibility

$n$ represents the number of independent experiments performed in parallel. Unpaired two-tailed Student's $t$-test was implemented for group comparisons as indicated in the figure legends. *$P < 0.05$; **$P < 0.01$; ***$P < 0.001$; ****$P < 0.0001$. Unless otherwise indicated, no statistical method was used to predetermine sample size. No data were excluded from the analyses; the experiments were not randomized; the Investigators were not blinded to allocation during experiments and outcome assessment.

### Reporting summary

Further information on research design is available in the Nature Portfolio Reporting Summary linked to this article.

## Data availability

All data and materials presented in this manuscript are available from the corresponding author (W.W.). Access to materials will be granted within two weeks of request submission, subject to completion of the MTA. The raw data of off-target analysis generated in this study have been deposited in the China National Center for Bioinformation-National Genomics Data Center database under accession code PRJCA025977. The structure data used in this study are available in the PDB database under accession code 1SSP Source data are provided with this paper.

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

## Acknowledgements

This project was supported by funds from the National Science Foundation of China (NSFC31930016), the Peking-Tsinghua Center for Life Sciences (to W.W.), the Fellowship of China Postdoctoral Science Foundation (8206400139 to Z.Y.), and Changping Laboratory (to W.W.).

## Author contributions

W.W. supervised this project. Z.Y., Xiaoxue Zhang, and X.W. conceptualized the idea. Z.Y., Xiaoxue Zhang, X.W., J.L., and W.W. designed the experiments. Z.Y., Xiaoxue Zhang, X.W., and J.L. conducted the experiments and analyzed the data with assistance from Xue Zhang, Y.Z., H.T., and X.C., J.R. analyzed high-throughput sequencing data. Y.Y. constructed high-throughput sequencing libraries. Z.Y., Xiaoxue Zhang, and W.W. wrote the manuscript with the help of all authors.

## Competing interests

W.W., Z. Y., Xiaoxue Zhang, and X.W. have submitted a patent application to the National Intellectual Property Administration, PRC patent office pertaining to the findings of this work (application number PCT/CN2024/092138). The applicants are Peking University and Changping Laboratory. W.W. is a scientific advisor and founder of EdiGene and Therorna. The remaining authors declare no competing interests.
