## [Peer Review File · Nature Communications]

Programmable DNA pyrimidine base editing via engineered uracil-DNA glycosylaseReviewers' Comments:

Reviewer #1:

Remarks to the Author:

The most important point in this revision was the comparison with base editors from two recent papers. For this comparison, the authors stated as follows.

"we selected five highest ranking sites from both articles, with Site 17-21 from the Nat. Biotechnol. article (PMID: 38168994) and Site 22-24 from the Mol. Cell. article (PMID: 38377993)."

For brevity, I will refer to the Nat. Biotechnol. article (PMID: 38168994) and Mol. Cell. article (PMID: 38377993) as the N and M papers, respectively.

I found that site 17 is not from the N paper. I also found that sites 19 and 20 have low editing efficiencies in the N paper, far from highest-ranking sites, and that sites 18 and 21 are medium efficiency sites in the N paper. Similarly, sites 24 is not from the M paper.

These discrepancies between the authors' response and the actual selection of target sequences make me suspect the fairness of the target site selection, which appears to be biased towards the authors' base editor, DrUNG mutant 6. Unless the authors test many more sites, including at least the five highest ranking sites from the N paper and five from the M paper, as I initially recommended, I do not think that the evidence supporting that DrUNG mutant 6 has a generally higher activity than recently developed competitor base editors is compelling.

I have also checked the responses to the comments of reviewer 2. Many of the comments by reviewer 2 have been addressed. However, the following comments need to be addressed.

Regarding comment 4 (Line 108: The claim that combining this system with preferential polymerases to achieve editing outcome specificity seems unfounded. Do you have data to support this claim?), the authors have not provided any data to support this claim.

Regarding comment 7 (An analysis of off-target editing should be provided for the reader. For example, analyzing Cas-dependent, Cas-independent, and RNA off-target sites for each evolved variant would be critical for understanding any unintended editing produced by the system), the authors have not determined any Cas-independent off-target effects, although reviewer 2 clearly mentioned it. Furthermore, the experiments appear to have been conducted with only one replicate.

Overall, the revised manuscript is still too preliminary to be published at Nature Communications.

Reviewer #2:

Remarks to the Author:

Using the previously submitted and reviewed manuscript as a reference, the authors have adequately fulfilled most of the concerns I brought up as a reviewer. I agree with publication.

Thank you very much to all the reviewers for their valuable comments. In response to Reviewer #1's suggestions, we have updated our manuscript. The revisions primarily focus on the following aspects:

1. We have compared our developed TBE, based on DrUNG, with other recently published base editors that utilize hUNG, in terms of their editing efficiencies. To ensure a fair comparison, we selected the top ten sites with the highest editing efficiencies from two studies (*Nat. Biotechnol.* (PMID: 38168994) and *Mol. Cell.* (PMID: 38377993)).
2. We conducted an off-target analysis for TBE that is independent of Cas9.
3. We explored overexpressing Pol η , a translesion synthesis (TLS) polymerase known for inserting specific bases opposite AP sites (PMID: 36624150).

Included below is a point-by-point response.

Reviewer Comments:

Reviewer #1 (Remarks to the Author):

The most important point in this revision was the comparison with base editors from two recent papers. For this comparison, the authors stated as follows.

“we selected five highest ranking sites from both articles, with Site 17-21 from the *Nat. Biotechnol.* article (PMID: 38168994) and Site 22-24 from the *Mol. Cell.* article (PMID: 38377993).”

For brevity, I will refer to the *Nat. Biotechnol.* article (PMID: 38168994) and *Mol. Cell.* article (PMID: 38377993) as the N and M papers, respectively.

I found that site 17 is not from the N paper. I also found that sites 19 and 20 have low editing efficiencies in the N paper, far from highest-ranking sites, and that sites 18 and 21 are medium efficiency sites in the N paper. Similarly, sites 24 is not from the M paper.

Response: Thank you very much for your suggestion. We apologize for the error made in selecting target sites during the initial revision process. We aimed to choose sites with the highest efficiency or those were frequently used in the main text of the two papers that discussed human Uracil-glycosylase (*Nat. Biotechnol.* (PMID: 38168994) and *Mol. Cell.* (PMID: 38377993)). However, while searching for raw data in Source Data Fig. 3 of the *Nat. Biotechnol.* paper, we mistakenly labeled the editing efficiency below, when it should have corresponded to the editing efficiency above. This error led to the selection of target sites with lower efficiency (Response Figure 1). Additionally, the target site that was incorrectly attributed to the *Nat. Biotechnol.* Paper actually originated from a pre-print on bioRxiv titled “Development of deaminase-free T-to-S base editor and C-to-G base editor by engineered human uracil DNA glycosylase” (<https://www.biorxiv.org/content/10.1101/2024.01.01.573809v1>), which also focuses on the development of thymine base editing. This contributed to confusion during our naming process.

	A			C			G		
T5	1.176599	1.101105	1.130029	1.355392	1.424804	1.361196	14.96982	14.20278	14.61849
T13	0.05153	0.051055	0.049493	0.202997	0.213168	0.206642	0.012102	0.013685	0.00975
T20	0.015225	0.014738	0.013367	0.105402	0.110532	0.111163	0.0203	0.017369	0.027095
Site T1	CACCTCAGCCCATCCCAGGT								
T5	6.230689	6.387375	6.307717	17.00091	17.60207	17.50256	14.04681	14.23984	14.36382
T11	0.588985	0.59082	0.534009	0.86996	0.96271	0.896264	0.474	0.488604	0.472928
T12	0.14435	0.147161	0.145749	0.305596	0.306646	0.242512	0.278561	0.271849	0.238883
T13	0.160281	0.141362	0.147563	0.315735	0.310996	0.298755	0.047795	0.053645	0.0381
Site T2	ACACTCACAGTTTCCAGACC								

Response Figure 1. Schematic representation of site selection mistakes, based on the original data from the *Nat. Biotechnol.* article.

These discrepancies between the authors’ response and the actual selection of target sequences make me suspect the fairness of the target site selection, which appears to be biased towards the authors’ base editor, DrUNG mutant 6. Unless the authors test many more sites, including at least the five

highest ranking sites from the N paper and five from the M paper, as I initially recommended, I do not think that the evidence supporting that DrUNG mutant 6 has a generally higher activity than recently developed competitor base editors is compelling.

Response: Thank you for your suggestion. In this revised version, we reselected sites with the highest editing efficiency from each of the two papers, choosing sites 31-40 from the *Nat. Biotechnol.* paper (PMID: 38168994) and sites 41-50 from the *Mol. Cell.* paper (PMID: 38377993). hUNG mutant 1 is from the *Nat. Biotechnol.* paper, while hUNG mutant 2 and 3 are from the *Mol. Cell.* paper. For a fair comparison, we replaced DrUNG mutant 6 with these three hUNG mutants in our construct. The results demonstrate that DrUNG mutant 6 exhibits higher editing efficiency at these sites (Response Figure 2). However, the performance of the human UNG-based tools did not match the levels reported in these two papers. We attribute this discrepancy to differences in experimental methods. The *Nat. Biotechnol.* and *Mol. Cell.* papers used puromycin to enrich transfected cells and cultured them for seven days post-transfection, whereas we did not perform enrichment and collected cells three days post-transfection. Nonetheless, all mutants were subjected to identical experimental conditions in our comparative analysis, ensuring fairness in the comparison.

Response Figure 2. DrUNG mutant 6 demonstrates superior editing efficiency. The editing efficiency across several endogenous sites using sgRNA and nCas9(D10A) fused to DrUNG mutant 6, hUNG mutants 1, 2, and 3 is presented. Data are shown as the mean \pm standard deviation from $n = 2$ independent biological replicates.

I have also checked the responses to the comments of reviewer 2. Many of the comments by reviewer 2 have been addressed. However, the following comments need to be addressed.

Regarding comment 4 (Line 108: The claim that combining this system with preferential polymerases to achieve editing outcome specificity seems unfounded. Do you have data to support this claim?), the authors have not provided any data to support this claim.

Response: Thank you very much for your suggestion. In a previous study, Tong *et al.* demonstrated that the editing bias could be shifted from A-to-Y to A-to-T by co-expressing the TLS polymerase Pol η in AYBEv3 (PMID: 36624150). Inspired by these findings, we also overexpressed Pol η in the TBE system and observed that T-to-A editing could be enhanced by 1.1 to 6 times. This indicates that

co-expressing polymerases can indeed influence the specificity of thymine conversion into targeted bases.

Response Figure 3. Comparison of editing purity at endogenous sites 8, 16, and 17 before and after the addition of polymerase. Data are shown as mean values from n = 3 independent biological replicates.

Regarding comment 7 (An analysis of off-target editing should be provided for the reader. For example, analyzing Cas-dependent, Cas-independent, and RNA off-target sites for each evolved variant would be critical for understanding any unintended editing produced by the system), the authors have not determined any Cas-independent off-target effects, although reviewer 2 clearly mentioned it.

Response: Thank you very much for your suggestion. To conduct a comprehensive evaluation of TBE specificity, we performed a Cas9-independent off-target analysis for TBE using an orthogonal R-loop assay (PMID:32046215). We co-transfected HEK293T cells with plasmids encoding a SpCas9-based TBE or ABE, a SpCas9 on-target guide RNA (sgRNA), a catalytically inactive *Staphylococcus aureus* Cas9 (dSaCas9), and a SaCas9 sgRNA targeting a genomic locus unrelated to the on-target site. The SaCas9 sgRNA targeting sites were consistent with those discussed in the *Nat. Biotechnol.* and *Mol. Cell.* papers.

Through deep sequencing of the SaCas9 sgRNA targeting sites, we observed no significant off-target effects for TBE, while ABE exhibited notable off-target activity, with the highest off-target efficiency reaching up to 26.7%.

Response Figure 4. Whole genome and transcriptome off-target assessment of TBEs. a, Schematic representation of the Cas9-independent off-target analysis using an orthogonal R-loop assay. b, Editing efficiency at

six genomic loci targeted by SaCas9 sgRNA, unrelated to the on-target site. Data are presented as mean \pm s.d. from $n = 3$ independent biological replicates.

Furthermore, the experiments appear to have been conducted with only one replicate. Overall, the revised manuscript is still too preliminary to be published at Nature Communications.

Response: Thank you very much for your suggestion. All off-target detection experiments were performed in three independent biological replicates. Figure 3 shows the average values of three biological replicates. We have included relevant descriptions in the corresponding figure legends.

Reviewers' Comments:

Reviewer #1:

Remarks to the Author:

I do not have any more concerns. I think that the paper is now appropriate for publication.